# Transcriptional Remodeling Patterns in Murine Dendritic Cells Infected with *Paracoccidioides brasiliensis*: More Is Not Necessarily Better

**DOI:** 10.3390/jof6040311

**Published:** 2020-11-24

**Authors:** Calliandra M. de-Souza-Silva, Fabián Andrés Hurtado, Aldo Henrique Tavares, Getúlio P. de Oliveira, Taina Raiol, Christiane Nishibe, Daniel Paiva Agustinho, Nalvo Franco Almeida, Maria Emília Machado Telles Walter, André Moraes Nicola, Anamélia Lorenzetti Bocca, Patrícia Albuquerque, Ildinete Silva-Pereira

**Affiliations:** 1Laboratory of Molecular Biology of Pathogenic Fungi, Department of Cell Biology, Institute of Biological Sciences, University of Brasília, Brasília, DF 70910-900, Brazil; cdssilva@gmail.com (C.M.d.-S.-S.); fahejml@gmail.com (F.A.H.); ildinetesp@gmail.com (I.S.-P.); 2Molecular Pathology Post-Graduation Program, University of Brasília Medical School, Brasília, DF 70910-900, Brazil; 3Faculty of Ceilândia, University of Brasília, Brasília, DF 72220-275, Brazil; atavares@unb.br; 4Division of Allergy and Inflammation, Department of Medicine, Beth Israel Deaconess Medical Center, Harvard Medical School, Boston, MA 02215, USA; junior.getulio@gmail.com; 5Fiocruz Brasília, Oswaldo Cruz Foundation, Brasília, DF 70904-130, Brazil; taina.raiol@fiocruz.br; 6Faculty of Computing, Federal University of Mato Grosso do Sul, Campo Grande, MS 79070-900, Brazil; cnishibe@gmail.com (C.N.); nalvojr@gmail.com (N.F.A.); 7Department of Molecular Microbiology, Washington University School of Medicine, St. Louis, MO 63110-1093, USA; daniel.molecular@gmail.com; 8Department of Computer Science, University of Brasília, Brasília, DF 70910-900, Brazil; mariaemilia@unb.br; 9Faculty of Medicine, University of Brasília, Brasília, DF 70910-900, Brazil; amnicola@unb.br; 10Laboratory of Applied Immunology, Department of Cell Biology, Institute of Biological Sciences, University of Brasília, Brasília, DF 70910-900, Brazil; albocca@unb.br

**Keywords:** *Paracoccidioides brasiliensis*, fungal innate immunity, dendritic cells, resistance, susceptibility, A/J and B10.A mouse strains, autophagy

## Abstract

Most people infected with the fungus *Paracoccidioides* spp. do not get sick, but approximately 5% develop paracoccidioidomycosis. Understanding how host immunity determinants influence disease development could lead to novel preventative or therapeutic strategies; hence, we used two mouse strains that are resistant (A/J) or susceptible (B10.A) to *P. brasiliensis* to study how dendritic cells (DCs) respond to the infection. RNA sequencing analysis showed that the susceptible strain DCs remodeled their transcriptomes much more intensely than those from the resistant strain, agreeing with a previous model of more intense innate immunity response in the susceptible strain. Contrastingly, these cells also repress genes/processes involved in antigen processing and presentation, such as lysosomal activity and autophagy. After the interaction with *P. brasiliensis*, both DCs and macrophages from the susceptible mouse reduced the autophagy marker LC3-II recruitment to the fungal phagosome compared to the resistant strain cells, confirming this pathway’s repression. These results suggest that impairment in antigen processing and presentation processes might be partially responsible for the inefficient activation of the adaptive immune response in this model.

## 1. Introduction

Paracoccidioidomycosis (PCM) is an endemic disease caused by the thermally dimorphic fungi *Paracoccidioides* spp. The disease is mainly found in humid tropical and subtropical areas of several Latin American countries, especially in Brazil, Colombia, Venezuela, Argentina, and Ecuador [1,2,3]. Although the incidence and prevalence are not fully known due to the noncompulsory nature of its notification, it is deemed the most prevalent systemic mycosis in Brazil, where 80% of all PCM cases are reported [3]. In immunocompetent hosts, PCM case fatality rates are usually less than 5%, but it is associated with high morbidity due to frequent chronic sequelae [4,5]. These morbidity rates are even higher in immunocompromised patients [5,6]. In Brazil, PCM accounted for approximately 51.2% (1853) of the total deaths attributed to the upper respiratory systemic fungal diseases between 1996 and 2006 [1,6].

Understanding the protective responses of the host’s immune system to fungal infections can help predict disease progression and might directly influence the patient’s treatment and prognosis. Resistance or susceptibility to *P. brasiliensis* infection is influenced by several factors, including fungal inoculum size and lineage, as well as the host’s age, genetic background, gender, overall health, the efficiency of antigen-presenting cells (APCs), such as dendritic cells (DCs), B-cells and differentiated macrophages, the infected site costimulatory microenvironment, and the type of CD4^+^ T helper cell (Th) induced [7,8,9,10]. The overall result of these differences and the immune response’s polarization pattern determine the PCM clinical form. A protective host defense mechanism is believed to be based on cell-mediated immunity with a predominant Th1 cytokine (INF-γ, IL12, IL2, and TNF-α) production resulting in classical macrophage activation that will kill or inhibit fungal growth [3,11,12,13,14]. Though, an increased regulatory T cell activity with excessive immune suppression (high levels of IL10 and TGF-β, soluble or membrane-associated with LAP-1, and high expression of CTLA-4/CD152) leads to the severe forms of the disease [14,15]. In general, a Th2 and Th9 response usually leads to an uncontrolled inflammatory process (acute form), whereas a deficient Th1 mixed with a Th17 immune response leads to the chronic form [11,12,13]. In this sense, the comparative analysis of the early immune response to a fungal pathogen employing animal models with different immune response profiles could bring new clues about host–pathogen interaction, the stabilization of immunological patterns, and disease progression.

Different PCM mammalian (e.g., murine, rat, guinea pigs, hamsters, and rabbits) and nonmammalian models (e.g., amoebas, nematode *Caenorhabditis elegans*, and insect *Galleria mellonella*) have been used to investigate distinct aspects of fungal infection and host–fungal interaction in a complex organism, such as fungal virulence, pathogenesis, immunological response, test pharmacological therapies, and find novel antimycotic compounds [13,16,17,18]. Albeit, the gold standard for in vivo studies still is the murine model of infection, and there are well-established models of PCM resistance (e.g., A/Sn or A/J murine strains) and susceptibility (e.g., B10.A, BIOD2/nSn, and BIOD2/oSn murine strains), both sharing high similarity to most common host responses observed in humans [19,20,21,22,23,24,25]. The B10.A, BIOD2/nSn, and BIOD2/oSn isogenic strains mimic the chronic, progressive, and disseminated forms of human PCM, whereas the A/Sn or A/J strains have similarities to the regressive or localized forms of infection [15,19,22]. The preeminent hypothesis in the resistance/susceptibility PCM model is that susceptibility to *P. brasiliensis* is associated with a stronger and more efficient initial innate immune response, which is later heavily repressed [26]. In contrast, resistance is associated with an initially milder/deficient response that later develops to a resistance pattern in the course of infection [26]. In the results previously reported by our group, this dichotomy was also found at the molecular level on GM-CSF- and M-CSF-induced bone marrow-derived macrophage from resistant (A/J) and susceptible (B10.A) mouse strains infected by *P. brasiliensis* [25].

Dendritic cells (DCs) play a pivotal role in the immune system as the most effective antigen-presenting cells and a mediator between innate and adaptive immune responses. Their potential for fine-tuning the host’s responses leading to the control or the eradication of the infection and its role in PCM has been highlighted in the literature [27,28,29,30,31]. Furthermore, although most in vivo studies focus on the late immune response presented by both resistant and susceptible mice, these studies suggested essential differences in the profile of pattern recognition receptors (PRRs) used by these hosts in their initial interaction with *P. brasiliensis* [19,22,26,27,32]. As the activation of different PRRs would result in different signaling pathways and immune responses, we decided to invest in a broader analysis of gene expression of bone marrow-derived DCs (BMDCs) from resistant and susceptible mice in response to *P. brasiliensis* infection. To achieve this, we have employed RNA sequencing (RNA-seq) to provide a global picture of early phase host gene expression in response to *P. brasiliensis* interaction. We found that BMDCs from the susceptible mouse presented a more intense response to infection, suggesting that an early immunological overreaction to this fungus might be linked to host susceptibility.

## 2. Materials and Methods

### 2.1. Fungal Cells and Growth Conditions

The virulent strain Pb18 of *P. brasiliensis* was maintained by weekly subcultivation in semisolid Fava-Netto’s medium at 37 °C and used in the experiments after 7 days of growth. Yeast cells were resuspended in PBS and adjusted to the desired concentration based on hemocytometer counts using the Janus Green B vital dye to determine viability [33]. Only cultures with viability greater than 90% were used in our experiments. The virulence of the strain was maintained by in vivo passages in mice every 3 months.

### 2.2. Mouse Strains and Bone Marrow-Derived Cells Differentiation

*P. brasiliensis*-resistant (A/J) and -susceptible (B10.A) male mice [19,22,26,34], between 6 and 12 weeks old, were obtained from the Immunology Department of the University of São Paulo Biomedical Sciences Institute, Brazil. The animals were housed with food and water ad libitum at the Animal Care Center of the Biological Institute of the University of Brasília, Brazil. The mice were euthanized, and their bone marrows collected. All procedures involving animals were performed following the animal use guidelines according to Brazilian laws and approved by the Committee on Ethical Use of Animals (Proc. UnB Doc 52657/2011).

Bone marrow-derived macrophages (BMMs) and dendritic cells (BMDCs) were generated from bone marrow cells, as previously described [35]. Briefly, 2 × 10^6^ bone marrow cells were plated on nontreated 100 mm culture dishes in complete RPMI-1640 medium (Sigma-Aldrich, Saint Louis, MO, USA) supplemented with 10% heat-inactivated fetal bovine serum (FBS; Thermo Fisher Scientific, Waltham, MA, USA), 50 µg/mL of gentamicin, 50 µM 2-mercaptoethanol (Sigma-Aldrich), and 20 ng/mL recombinant GM-CSF (PeproTech, Ribeirão Preto, SP, Brazil). The cultures were incubated for 8 days at 37 °C in a humidified 5% CO_2_ atmosphere. On the third day, 10 mL of fresh completed medium was added to the culture. Half of the plated medium was removed on the sixth day and supplemented with fresh complete medium. Nonadherent BMDCs in the culture supernatant and loosely adherent cells were harvested by gentle washing on the eighth day. We typically obtain around 80% of these cells expressing MHC class II and CD11c, which characterize bone marrow-derived DCs [36,37]. The attached BMMs were detached from plates with TrypLE™ Express (Thermo Fisher Scientific) and were separately collected.

### 2.3. Ex Vivo Infection of Dendritic Cells from PCM-Resistant and -Susceptible Mouse Strains

BMDCs uninfected (control) and infected with *P. brasiliensis* at a multiplicity of infection (MOI) of 5:1 were incubated in RPMI 10% fetal bovine serum for 6 h in a humidified atmosphere of 5% CO_2_ at 37 °C. This MOI has been previously shown to be nondeleterious to macrophage cultures [38,39]. After the incubation time, the culture supernatants were collected for cytokine and chemokine measurements, and the BMDCs were lysed for total RNA extraction.

### 2.4. Cytokine and Chemokine Measurements

The levels of the cytokines TNF-α, IL6, IL10, and the chemokine MCP-1 in the coculture supernatants were quantified by a capture enzyme-linked immunosorbent assay (ELISA) using specific kits from ELISA Ready-SET-Go!^®^ (eBioscience, San Diego, CA, USA), according to the manufacturer’s instructions. The absorbance values were measured in a spectrophotometer (SpectraMax M5—Molecular Devices, San Jose, CA, USA) and analyzed with the SoftMax 5.2 software. Cytokine and chemokine concentrations were determined using a standard curve, following the kit’s recommendations.

### 2.5. Ex Vivo Infection of Resistant and Susceptible BMMs and BMDCs with P. brasiliensis for LC3 Immunofluorescence

Eight hundred thousand (8 × 10^5^) BMMs or BMDCs were plated onto MatTek^®^ glass-bottom dishes for 24 h (MatTek, Ashland, MA, USA). On the day of the interaction, cells from 5-day-old cultures of *P*. *brasiliensis* were collected, vortexed in PBS, and then passed through a 40 µm cell strainer before counting in hematocytometer. Following that, fungal cells were inoculated on the MatTek^®^ Petri dishes containing BMMs or BMDCs at a MOI of 1:1. The dishes were incubated for 12 h at 37 °C in the presence of 5% CO_2_ to allow infection. After infection, MatTek^®^ dishes were used in immunofluorescence experiments to locate the microtubule-associated protein 1A/1B-light chain 3 (LC3), widely used to monitor autophagy [40].

### 2.6. Immunolocalization of LC3 in Infected BMMs and BMDCs

After 12 h of infection, the cells were fixed with ice-cold methanol for 10 min and washed with PBS. Afterward, they were incubated with primary antibody (rabbit polyclonal IgG against human LC3, 1:1000 dilution (Santa Cruz Biotechnology, Dallas, TX, USA) for 1 h at 37 °C. Subsequently, the cells were washed three times with PBS and incubated with a secondary antibody (goat IgG against rabbit IGG conjugated with Alexa Fluor^®^ 488, dilution 1: 2000 (Thermo Fisher Scientific) for 1 h at 37 °C. Following secondary antibody incubation, the cells were washed three times with PBS, and MatTek^®^ dishes were mounted with ProLong Gold Antifade Mountant (Thermo Fisher Scientific) and a microscope coverslip. MatTek^®^ dishes were then observed by epifluorescence microscopy on a Zeiss Axio Observer Z1 equipped with a 63x NA 1.4 objective (Zeiss, Oberkochen Germany). Micrographs were recorded with a cooled CCD camera and the Zeiss ZEN software. Image stacks were deconvolved with a constrained iterative algorithm on Zeiss Zen and manipulated on Adobe Photoshop (Adobe, Mountain View, CA, USA). No nonlinear modifications were made to the images, and when brightness adjustments were made, they were applied uniformly to the entire image. To quantify LAP. The phagocytosed fungal cells in each field were counted as well as the number of phagocytosed fungal cells positive for LC3. The analysts who performed the counting were blinded to the identity of the samples. The percentage of LC3-associated phagocytosis (LAP) was measured by the number of phagocytosed fungal cells positive for LC3 divided by the total number of phagocytosed fungal cells. As positive control, we made in parallel similar experiments using BALB/c BMMs infected with *C. neoformans* or *C. albicans* (Appendix A), a condition in which LAP has been previously detected [40].

### 2.7. RNA Isolation

After 6 h of interaction, the total RNA of *P. brasiliensis*-infected dendritic cells or uninfected control cells was isolated using RNeasy^®^ Plus Mini Kit (QIAGEN, Hilden, Germany), following the manufacturer’s protocol. After the quality analysis (Bioanalyzer 2100—Agilent, Santa Clara, CA, USA) and quantification (Qubit^®^ 2.0 Fluorometer—Life Technologies, Carlsbad, CA, USA), 3.5–6.0 μg of total RNA from BMDCs of the different experimental conditions was prepared for transport at room temperature employing the RNA stable kit (Biomatrica, San Diego, CA, USA), according to the manufacturer’s recommendations.

### 2.8. Sequencing Parameters

Twelve barcoded libraries after poly(A) + RNA selection, according to the Illumina TruSeq RNA-seq methodology, were sequenced from total BMDC RNA samples obtained from resistant (A/J) and susceptible (B10.A) strains, infected or not by Pb18. Three noninfected and three Pb18-infected BMDC samples were used from each mouse strain. High-throughput RNA sequencing (RNA-seq) was subsequently performed using the Illumina HiSeq 2000 Analyzer platform at the Scripps DNA Sequencing Facility, generating paired-end reads of 100 base pairs (bp).

### 2.9. RNA-Seq Data Analysis

The high throughput pipeline analysis of both transcriptomes of BMDCs from these two mice strains infected or not with the virulent Pb18 isolate of *P. brasiliensis* was analyzed as described below. FASTQ files obtained from Illumina sequencing [41] were evaluated using the FastQC software [42]. The adapters identified by FastQC were removed using the Cutadapt software [43], while the low-quality sequences were filtered using PRINSEQ [44]. Qualified reads were mapped onto the mouse reference genome (mouse build mm10) using TopHat2 [45]. The final BAM mapping files were ordered and indexed using SAMtools [46], followed by a read count using HTSeq-count [47]. Transcripts with low counts (CPM < 1) and not present in at least two libraries were removed from the analysis.

The differentially expressed genes (DEG) were estimated using the edgeR library, applying TMM library size normalization and likelihood ratio test, implemented in software R version 3.6.1 [48]. The false discovery rate (FDR) was controlled by the Benjamini-Hochberg algorithm [49]. A fold change ≥ ± 1.4 and an FDR-adjusted *p*-value < 0.05 were used as cutoff criteria for differential gene expression.

Genes considered differentially expressed were used for the Gene Ontology (GO) and KEGG enrichment analysis by ClusterProfiler [50]. Only nodes with FDR-adjusted *p*-values < 0.01 and *p*-values < 0.05 were considered statistically significant for GO and KEGG analysis, respectively. Simplify method was used to reduce the enriched GO terms redundancy. Plots were produced in R using ggplot2. Heatmaps were generated using the ComplexHeatmap library and the Gene Ontology database [51].

### 2.10. Data Access

All sequencing data were deposited in NCBI’s Gene Expression Omnibus (GEO) database under accession number GSE158289.

### 2.11. RNA-Seq Validation by Quantitative PCR (RT-qPCR)

We performed RT-qPCR of nine genes (Appendix A) related to innate immunity using the same samples of total BMDC RNA used in the RNA-seq. After DNase I treatment (included in the RNeasy^®^ Mini Kit Plus), first-strand cDNAs were synthesized from 500 ng of total RNA for each sample following the instructions for SuperScript III (Thermo Fisher Scientific). The RT-qPCR was performed using SYBR Green Master Mix (Thermo Fisher Scientific) with this dye’s standard cycling condition. Gene expression changes relative to control were estimated using the 2^−∆∆Ct^ method [52].

The internal control used was the 40S ribosomal protein S9 (RPS9) gene (Appendix A), as described previously by our group [36,39]. Specific oligonucleotides for the genes encoding MyD88, NF-κB, TNF-α, and IL1β were designed as described before [39] and based on sequences obtained from the mouse transcriptome database (http://www.informatics.jax.org). The oligonucleotide sequences for the genes encoding IL6, IL10, IL12αp35, CXCL10, and CCL22 were obtained from the PrimerBank database http://pga.mgh.harvard.edu/primerbank/ [53]. All primer sequences are listed in Appendix A.

### 2.12. Statistical Analysis

Three independent experiments were performed for every outcome measured. The differences between the groups, for all experiments except RNA-seq, were analyzed by Student’s *t*-test, two-way ANOVA with Tukey’s multiple comparisons post-test, or by Fisher’s exact test. The statistical analysis was performed using GraphPad Prism 7.0.0 (GraphPad Software, San Diego, CA, USA). A *p*-value < 0.05 was considered significant.

## 3. Results

### 3.1. P. brasiliensis Infection Triggers Widespread Transcriptional Remodeling in a PCM-Susceptible Mouse Strain

We characterized the early transcriptional response of BMDCs derived from well-established murine models of resistance (A/J) and susceptibility (B10.A) to PCM after their interaction with *P. brasiliensis* [15,22,23]. For that, BMDCs were coincubated with or without *P. brasiliensis* cells at a MOI of 5:1 for 6 h; afterward, these cells were collected for RNA extraction and sequencing.

For all the samples, each replicate yielded an average of 16 million reads after the sequencing of both transcriptomes, and all the analyzed samples had more than 97% of their reads mapped to the mouse reference genome (Appendix A). To generate a set of differentially expressed genes in *P*. *brasiliensis*-infected BMDCs relative to uninfected control, for both mouse strains, we adopted statistical and biological significance thresholds of adjusted *p*-value < 0.05 and FC ≥ ±1.4, respectively.

We observed a significant disparity in the number of differentially expressed genes upon infection (red dots) between cells derived from both mouse strains, as presented in Figure 1A. BMDCs from the susceptible strain modulated a much higher number of genes (2278) upon infection with *P. brasiliensis* than BMDCs from the resistant strain (221), and both sets have few genes in common (189) (Figure 1B). In fact, the resistant BMDCs seem to hardly alter their noninfected transcript landscape following *P. brasiliensis* infection. This difference was also reflected in the general immune response of both mouse strains upon infection with *P. brasiliensis* and could be more clearly discerned by analyzing the number of differentially expressed genes (DEGs) clustered in the Gene Ontology (GO) terms: immune response, immune system process, inflammatory response, innate immune response, and metabolic process genes. As presented in Figure 1B,C, BMDCs from the resistant A/J strain had only 213 upregulated genes and eight downregulated ones. In contrast, BMDCs from the susceptible B10.A strain not only presented a significantly higher overall number of upregulated DEGs (1128) but also displayed 1150 downregulated DEGs and a higher number of DEGs clustered in those GO terms than in the resistant strain. It was also interesting to note that the susceptible mouse strain seems to downregulate its metabolic processes in response to infection. The complete list of genes is presented in Appendix A.

The RNA-seq results were validated using RT-qPCR to assess transcript levels of several genes related to the innate-immune response; namely, those encoding the cytokines TNF-α, IL1β, IL6, IL10, and IL12, the chemokines CCL22 and CXCL10, the molecular adapter MyD88, and the transcription factor NF-κB (Appendix A). Comparing infected to noninfected cells, a similar increase in TNF-α, IL6, and IL10 transcripts accumulation and cytokines release was observed for both strains (Appendix A). However, the susceptible mice produced significantly higher levels of those cytokines in comparison to the resistant strain. The MCP-1 (CCL2) release was also stimulated after infection in both mouse strain, in a similar fashion (Appendix A). Nevertheless, the MCP1-transcript was not differentially expressed after infection in either mouse strain, which could be explained by the post-transcriptional regulation of MCP-1 mRNA stability [54,55]. Depending on the stimulus, MCP1-transcript accumulation peak tends to occur at 2 h returning to normal levels at 4 h [56,57]; in our analyses, both RNA and cytokine/chemokine samples were collected after 6 h of infection with *P. brasiliensis*.

### 3.2. PCM-Resistant Mouse Strain Reveals a Precise and Coordinated Immune Response upon P. brasiliensis Infection

A selected global GO analysis profile for the DEGs in BMDCs from resistant and susceptible strains upon infection with *P. brasiliensis* is represented in Figure 2. Genes modulated by BMDCs from the susceptible strain significantly clustered in 98 GO terms, of which 12 were downregulated (Appendix A). Meanwhile, modulated genes from BMDCs of the resistant strain were significantly grouped in 81 GO terms, and all were upregulated (Appendix A). Most of the GO biological processes, in both mouse strains, were involved in innate immunity, its regulation, or in immunity against internalized pathogens (Figure 2).

Further analysis focusing on GO biological processes revealed enrichment of upregulated genes in GO immunological processes categories (*p* < 0.01) in both resistant and susceptible strains (Appendix A). The resistant BMDCs display fewer DEGs, grouped in a smaller number of categories but displaying a higher number of upregulated genes, in an apparent higher interconnected organization (Figure 3), which might reflect a more precise and coordinated response to infection than the susceptible BMDCs (Figure 4). Both models shared six upregulated biological process categories: response to interferon-beta, cytokine secretion, chemotaxis, regulation of cell killing, positive regulation of phagocytosis, and production of molecular mediators of the immune response (Figure 2 and Figure 3). However, they markedly differed in the modulation of some categories. For example, BMDCs from the resistant strain upregulated genes related to inflammatory response, positive regulation of cytokine production, neutrophil chemotaxis, and positive regulation of the apoptotic process (Figure 2). In contrast, BMDCs from the susceptible strain upregulated genes involved in macrophage migration, negative regulation of catalytic activity, innate immunity, and antigen processing and presentation (Figure 2).

GO analysis performed for the downregulated genes of BMDCs from the susceptible strain revealed enrichment for categories related to catabolic processes, mainly involving lipid and small molecule metabolism, and lyase activity, as well as coenzyme binding, which are in agreement with the previously observed upregulation of genes involved in negative regulation of catalytic activity (Figure 2 and Appendix A). We did not observe any significant enrichment (adjusted *p* < 0.01) of the immunological process among the repressed genes from the resistant strain BMDCs (Figure 2 and Appendix A).

In terms of Kyoto Encyclopedia of Genes and Genomes (KEGG) pathway analysis, a total of 69 upregulated pathways clustered in the susceptible strain and 47 in the resistant strain (Appendix A), with both strains similarly clustering upregulated genes in 14 pathways related to immune system processes, but differing only in their enrichment levels and genes counts (Figure 5).

Among the differences in gene modulation upon infection between the two strains, we noticed a positive regulation of apoptotic and the leukocyte apoptotic processes only in the resistant model (Figure 2). In contrast, the susceptible strain presented an upregulation of several signal transduction pathways, including pathways involved in DC maturation, adaptive immune response polarization, apoptosis, and autophagy such as the MAPK, HIF-1, and PI3K-Akt (Figure 5) [58,59,60,61].

Similarly to the GO terms, only BMDCs of B10.A strain displayed downregulation of KEGG pathways, most related to catabolism. The peroxisome proliferator-activated receptor (PPAR) signaling pathway, an essential modulator of the immune response, was also downregulated in this mouse model (Figure 5). 

### 3.3. Resistant and Susceptible Strains had Significant Differences in the Modulation of Genes Related to Antigen Presentation, Autophagy, and Lysosome Function

Comparing the transcriptomes of *P. brasiliensis*-infected resistant and susceptible BMDCs, we noticed some critical differences in the GO and KEGG pathway analysis between the two mouse strains. The modulation of genes involved in autophagy (GO:0006914), lysosome (GO:0005764), and antigen processing and presentation (GO:0019882) are highly interconnected and have critical elements repressed in the susceptible strain, as seen in Figure 6, indicating a decreased functionality of these pathways (Figure 2 and Figure 5). It is interesting to note that some genes from autophagy, lysosome, and antigen processing and presentation ontologies are similarly upregulated in both strains (Figure 6). However, different from the resistant mice, in which the downregulated genes did not reach the cutoff limit, the susceptible mice modulated key genes leading to a repression of those pathways; e.g., the upregulation of the transcription factor Hif1α (hypoxia-inducible factor 1α), and the repression of catabolic enzymes, related to different exocytosis and endocytosis processes, regulatory enzymes, and pH altering enzymes, such as SGSH (N-sulfoglucosamine sulfohydrolase), SMPD1 (sphingomyelin phosphodiesterase 1), ACP5 (acid phosphatase 5), PLA2G15 (lysosomal phospholipase A2 group XV), PIK3R2 (phosphoinositide-3-kinase regulatory subunit 2), and ATP6V0D2 (ATPase H+ transporting V0 subunit D2) (Figure 6 and Appendix A).

There was also downregulation of critical transcriptional factors, such as TFEB (transcription factor EB—master regulator of lysosomal biogenesis, autophagy, lysosomal exocytosis, lipid catabolism, energy metabolism, and immune response), proteins such as Deptor (DEP domain-containing mTOR-interacting protein—mTOR inhibitor), Stx17 (syntaxin 17—SNARE essential for fusion of cellular membranes), Atg14 (autophagy related 14—determines the autophagy-specific PI3-kinase complex PI3KC3-C1 localization), Snx14 (sorting nexin 14—intracellular trafficking and required for autophagosome clearance), and receptors, such as TLR9 (intracellular DNA recognition), H2-DMa (a subunit of a heterodimeric H2-DM chaperone molecule), Fcgrt (Fc fragment of IgG receptor and transporter), HFE (homeostatic iron regulator—membrane protein similar to MHC class I-type proteins and associates with beta2-microglobulin (beta2M)), and NBR1 (NBR1 autophagy cargo receptor) (Figure 6 and Appendix A).

Considering processes related to antigen processing and presentation, one of the main DCs activities, most components of MHC (major histocompatibility complex) class I and MHC class II components were not significantly altered in the resistant mice (Figure 6C and Appendix A). Notwithstanding, we observed an enrichment in GO terms related to antigen cross-presentation by those DCs, such as the “antigen processing and presentation of endogenous peptide antigen via MHC class I/via ER pathway, TAP-independent” (Figure 2), and “antigen processing and presentation of endogenous peptide antigen via MHC class Ib” (Appendix A). In contrast, in BMDCS from the susceptible mice, there was enrichment in several components of GO terms and KEGG pathways related to antigen processing and presentation (Figure 2 and Figure 5), including the ones observed in the resistant strain, but also of other components of MHC class I, such as “MHC class I peptide loading complex” and “TAP binding” followed by downregulation of some MHC II components such as H2-DMa (MHC-IIb). This chaperone is critical for the release of class II HLA-associated invariant chain-derived peptides (CLIP) from the MHC II groove, freeing the peptide binding site, which might compromise MHC class II availability in the susceptible mouse [62] (Figure 6C and Appendix A).

### 3.4. PCM-Susceptible Mouse Strain Shows a Deficiency in Performing LC3-Associated Phagocytosis of P. brasiliensis

The susceptible strain transcriptional profile suggested a possible impairment of autophagy in the mouse model. Given the role of autophagy in the immune response against several microbes and our work regarding macrophages after the interaction with several fungal pathogens [40,63,64,65], including *P*. *brasiliensis* [66], experiments were performed to assess differences in LC3-associated phagocytosis (LAP) after *P. brasiliensis* infection between the two mouse strains. For this, we infected murine BMMs and BMDCs with the virulent strain Pb18, and after 12 h of interaction, we performed immunofluorescence experiments with antibodies to LC3, an autophagosome marker, which is also a marker for LAP [67] (Figure 7A). We observed a significant difference in LAP induction after *P. brasiliensis* infection between both macrophages and DCs derived from A/J and B10.A mouse strains (*p* < 0.0001). The percentage for LAP-positive cells decreased from 10.4% in resistant BMMs to 4.6% in susceptible BMMs, while the percentage of LAP-positive cells decreased from 14.1% in resistant BMDCs to 7.4% in susceptible BMDCs (Figure 7B). These differences suggest a possible link between the macrophages and dendritic cells’ ability to perform LC3-associated phagocytosis and susceptibility/resistance to *P. brasiliensis* infection.

## 4. Discussion

In several systemic mycoses, host resistance is associated with cellular immunity and proper activation of phagocytes, while susceptibility is associated with polarization towards type-2 immunity (Th2/Th9) and a marked impairment/depression of cellular-mediated immunity [15,68]. In the murine PCM model, the A/J (resistant) strain portrays an initial more controlled (mild) response to infection that evolves for activation of cellular immunity and phagocytes, resulting in a limited number of well-organized granulomatous lesions that evolve into self-healing, abundant neutrophil infiltration, and fungal destruction [15,20,26,68]. Meanwhile, B10.A (susceptible) shows greater activation of the innate immune response, leading to excessive NO secretion, which might lead to deletion and anergy of CD4^+^ cells, defective activation of cellular immunity, culminating in disseminated nonorganized inflammatory lesions containing high fungal loads [15,20,26,68,69]. Considering those differences, we decided to further investigate possible differences in the global transcriptional profile of BMDCs from both strains, focusing on their innate immune response against *P. brasiliensis* infection after 6 h of interaction.

We observed that BMDCs from the susceptible mouse presented a more intense and apparently disorganized gene modulation in response to infection in comparison to cells from the resistant mouse. Overall, the disparity in the numbers and terms of GO processes and KEGG pathways enriched upon infection in both groups agree with previous models of PCM resistance/susceptibility [15,69]. Although A/J and B10.A clustered genes in similar categories when using the immune system process GO database, the BMDCs in PCM-resistant mice (A/J) are probably mounting a more controlled and precise response, up-regulating monocyte’s and neutrophil’s recruitment, apoptotic process, cell killing, response to interferon-beta, and type I interferon and cytokine production. In comparison, PCM-susceptible mice (B10.A) induces an inadequate and disproportionate response by direct or indirectly downregulating several catabolic processes, essential for lysosomal function, and possibly to antigen presentation and the PPAR pathway, important in the modulation of inflammatory processes [70,71], while upregulating macrophage migration but not neutrophil or monocyte recruitment. 

Our transcriptomic results revealed that BMDCs from the resistant mice strain induce gene expression that, according to the literature, reinforces the idea of differential migration of neutrophils, while BMDCs from the susceptible mice strain induce macrophage migration during early interaction with *P. brasiliensis*. Neutrophils are seen in lesions of PCM patients and experimentally infected mice; and, when appropriately activated (IFN-γ, TNF-α, GM-CSF, and IL15), they are able to limit infection and fungal burden and are important sources of INF-γ and IL-17, especially at early stages of *P. brasiliensis* infection [72,73,74]. Neutrophil depletion at chronic PCM phases might attenuate lung fibrosis and inflammation. However, the absence of those cells at the initial acute phase of PCM exacerbates the inflammatory response indicating an important role of neutrophils in the early response to *P. brasiliensis* infection [75,76].

On the other hand, despite their role in resistance to *P. brasiliensis* infection, confirmed both in susceptible and resistant models of infection [77], macrophages can also be a relevant site of fungal replication and dissemination. Nonactivated alveolar macrophages, despite their ability to internalize yeasts both in vivo and in vitro, are frequently permissive to the multiplication of *P. brasiliensis,* while their activation by INFγ enhances their microbicidal activity [78]. In conclusion, both phagocytes play vital roles in the immune response against *P. brasiliensis*, albeit in different moments of the host–pathogen interaction. 

Another critical difference observed between BMDCs from susceptible and resistant mouse models of PCM relies on the modulation of genes related to antigen processing and presentation, a key function of DCs. Recognition, processing, and presentation of antigens by these cells determine adaptative response polarization and ultimately define the outcomes of host–pathogen interaction [7]. The differences in antigen processing and presentation in APCs is dependent, among other factors, on the rate of lysosomal proteolysis and the proper selection of epitopes [79]. Most genes of MHC class I and MHC class II components were not significantly modulated in the resistant mice. In contrast, BMDCs from the susceptible mouse had a broader enrichment in several GO term and KEGG pathway components related to antigen processing and presentation, especially in the MHC class I components; this seems to be in agreement with a disproportional early inflammatory profile of the susceptible mouse. 

BMDCs from the susceptible strain also displayed enrichment of GO and KEGG categories related to the repression of different lysosomal pathway elements. This catabolic repression in the susceptible strain might impact several processes related to fungal destruction and development of protective adaptive immune response, such as phagosomal activity, reactive oxygen species production, antigen presentation, and autophagy [62]. Taken together, these features indicate that fungal cells might remain longer inside DCs, avoiding effector functions of the immune system in the susceptible mice. In agreement with this hypothesis, Ferreira et al. (2007) had previously described that DCs from PCM-susceptible mice had not only a higher phagocytic index but also higher fungal viability after the interaction with *P. brasiliensis* than cells from the PCM-resistant mouse model [8].

Another transcriptional difference between BMDCs from both mouse models was the regulation of autophagy, a process closely related to several components of immunity to infection, including microbial killing, antigen presentation, and inflammation [80]. In the last two decades, several groups have described how autophagy can participate in innate and adaptative responses to different microbes. Furthermore, some pathogens developed ways to manipulate host autophagy for their benefit. LC3-associated phagocytosis (LAP) is a noncanonical form of autophagy, triggered by the engagement of surface recognition receptors, and a link between phagocytosis and the autophagy machinery. This process impacts immune activation and inflammatory response, and it is believed to be a safe pathway to control the lysosomal degradation of microbial pathogens [81,82,83]. Despite the lack of a double membrane autophagosome, LAP shares several of the canonical autophagy components, including Beclin1, various Atg proteins, the PI3K complex, and the soluble LC3-I conversion into the membrane-bound LC3-II [81]. So far, LAP has been shown to play a role in the antifungal immunity to several fungal pathogens, including *Aspergillus fumigatus*, *Candida* spp., *Cryptococcus neoformans*, and *Histoplasma capsulatum* [40,63,64,65]; our group has also been involved in the characterization of this process in the interaction of *P. brasiliensis* and macrophages in the last few years [66].

In DCs, antigen processing via autophagy modulates T cell immunity by promoting both endogenous and exogenous antigen presentation. This process is believed to be the primary intracellular pathway involved in the non-conventional endogenous antigenic peptide presentation in MHC class II, making it important in DCs for both self and foreign antigens presentation [62,84]. Autophagy also contributes to the DCs regulation of cytokine production and cell death. Both autophagy proteins and NADPH oxidase 2 (NOX2) are required for LC3-associated phagocytosis (LAP), stabilize the phagosome, and are crucial for the efficiency of MHC class II presentation of extracellular antigens [84]. We observed the repression of several key autophagy genes in susceptible mice, and this autophagic function repression was confirmed by the reduced percentage of LC3-II recruitment in the phagosomes of B10.A DCs and macrophages infected with *P. brasiliensis* in comparison to those from A/J mice.

Among the differences in the autophagy/LAP regulation, there was a significant downregulation of Deptor transcript and the upregulation of Hif1α transcript after infection in BMDCs from the susceptible strain. Deptor is an inhibitor of mTORC activity and, consequently, an activator of autophagy. In multiple human myeloma cells, the knockdown of Deptor was shown to trigger apoptosis and suppress autophagy [85]. The transcription factor Hif1α is a major regulator of innate immunity against pathogens and macrophage INF-γ-dependent control of infection [86,87]. In the interaction of macrophages with *H. capsulatum*, Hif1α was shown to limit fungal intracellular survival by reducing the recruitment of LC3-II to the fungal phagosome. In this case, *H. capsulatum* exploits host autophagy to survive [87,88]. Contrastingly, results from a parallel project from our group suggest that, in the interaction of different macrophages with *P. brasiliensis*, LAP is detrimental for the fungus, reinforcing this process’s potential role in macrophage ability to deal with this pathogen. In addition to its role in autophagy, Hif1α upregulates inducible nitric oxide synthase (iNOS). In our experiments, BMDCs from both mouse models upregulated the NOS2 gene (nitric oxide synthase 2, inducible); however, the upregulation in the susceptible mouse was significantly higher, and excessive NO production by the susceptible mouse was suggested to have a suppressive effect on T lymphocytes activation [69]. Our results suggest that the upregulation of Hif1α might be involved in the lower recruitment of LC3-II to the phagosome observed in the immunofluorescence assays and also in the increased accumulation of the iNOS transcript observed in BMDCs from the susceptible mouse.

In conclusion, our results corroborate the previously proposed intense activation of the inflammatory response in the susceptible PCM mouse model after infection with *P. brasiliensis*. In addition, we propose that suppression of highly interconnected processes, such as repression of lysosomal acidification, catalytic activity, and autophagy function, might negatively impact antigen processing and presentation by BMDCs from the susceptible mice leading to ineffective activation of the adaptive immune response and susceptibility to this fungal infection (Figure 8). What makes a host susceptible or resistant to infection is a crucial question for most infectious diseases, and many factors have been implicated in disease development, such as sex, nutritional status, smoking habits, pollution, and genetics [89]. Our work reinforces the significance of host genetic background in susceptibility to fungal infections. These findings might help in development of strategies to prevent or treat PCM, such as the use of DC biomarker detection to predict people with higher risks of developing the disease, information with significant prognostic impact. Therefore, further in-depth exploration and assessment of these processes, e.g., in vivo, with other cells or different time points of interaction, should help deepen the comprehension of the molecular mechanisms behind susceptibility/resistance not only for this neglected fungal infection but also for other infectious diseases.

## Figures and Tables

**Figure 1 jof-06-00311-f001:**
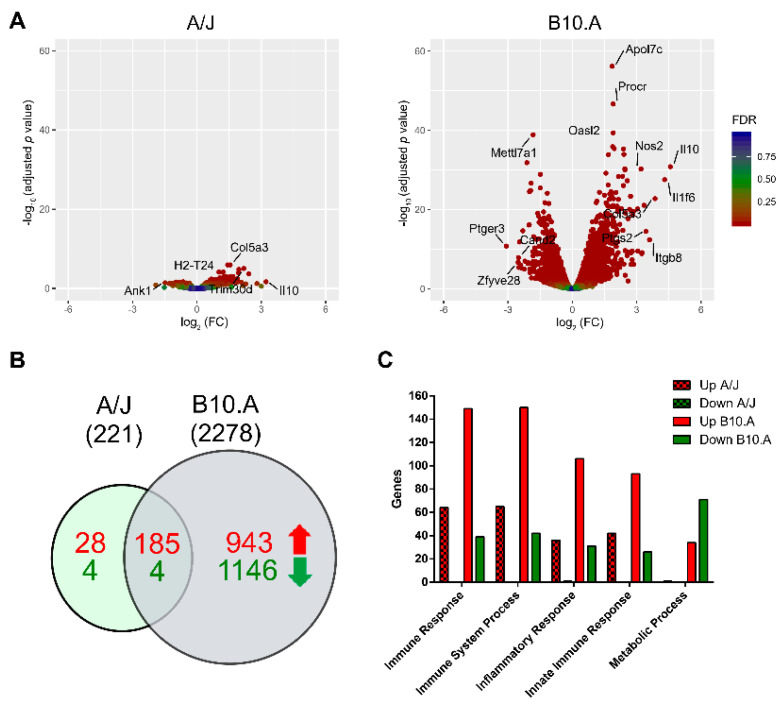
Differential gene expression of bone marrow-derived dendritic cells (BMDCs) from paracoccidioidomycosis (PCM)-resistant and -susceptible strains upon *P. brasiliensis* infection. (**A**) Volcano plot showing gene expression changes comparing *P. brasiliensis*-infected BMDCs derived from the resistant A/J (left) and the susceptible B10.A (right) mouse strains vs. control (noninfected) samples. The *y*-axis represents the -log_10_ values of the adjusted *p*-value, and the *x*-axis represents the log_2_ values of the fold change observed for each transcript. The top differentially expressed genes (DEGs: adjusted *p*-value < 0.05 and fold change ≥ ± 1.4) are indicated. (**B**) Venn diagram of positively (red) and negatively (green) regulated genes of A/J and B10.A mice strains. (**C**) Number of DEGs correlated to different Gene Ontology (GO) terms in A/J and B10.A mouse strains infected with *P. brasiliensis*.

**Figure 2 jof-06-00311-f002:**
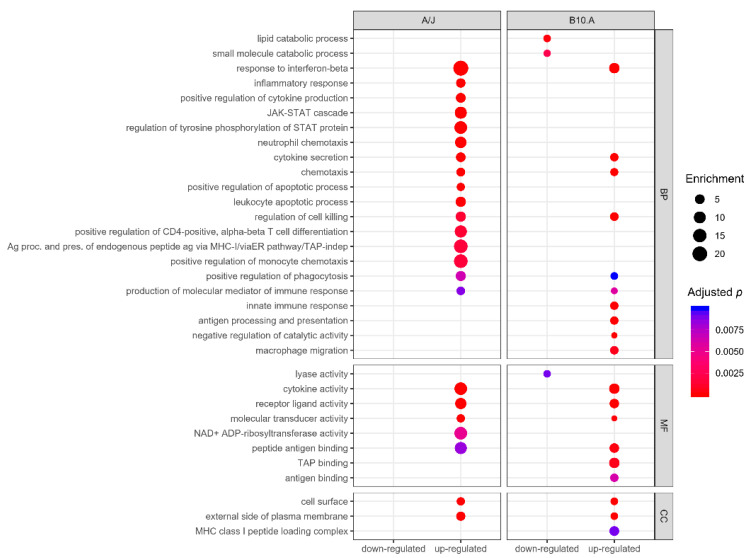
Gene ontology enrichment of differentially expressed genes in BMDCs after *P. brasiliensis* infection. Enriched ontological categories (adjusted *p*-value < 0.01) associated with up- or downregulated DEG in BMDCs derived from the resistant A/J (**left**) and the susceptible B10.A (**right**) mice strains. Dot size represents the enrichment (gene modulated ratio/gene background ratio) for each GO term. BP: biological process, MF: molecular function, CC: cellular component.

**Figure 3 jof-06-00311-f003:**
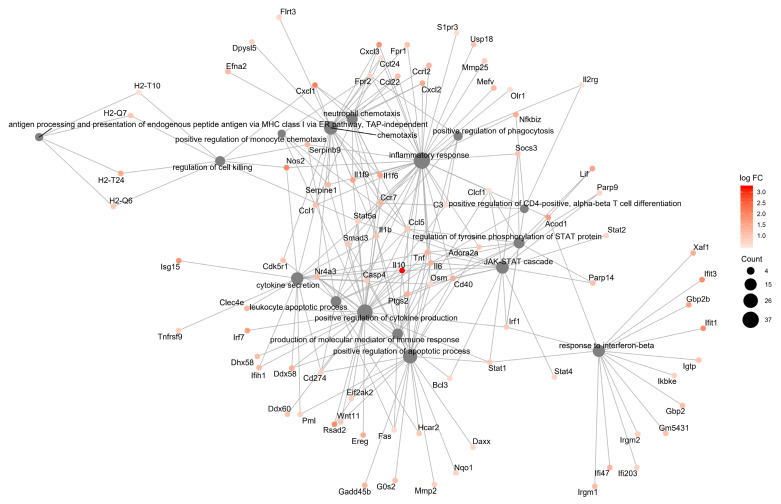
Functional interaction networks related to biological process Gene Ontology terms enriched by genes upregulated in response to *P. brasiliensis* infection in BMDCs from the resistant mouse strain, A/J. ClusterProfiler was used to generate the functional interaction networks formed by upregulated genes related to the GO Biological Process terms. Dot size represents the number of genes in each GO term.

**Figure 4 jof-06-00311-f004:**
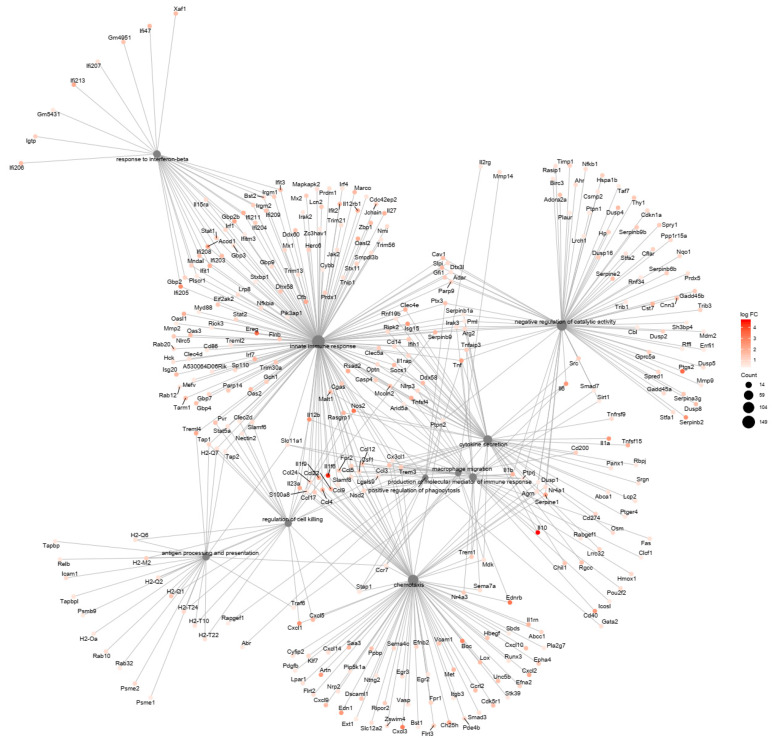
Functional interaction networks related to biological process Gene Ontology terms enriched by genes upregulated in response to *P. brasiliensis* infection in BMDCs from the susceptible mouse strain, B10.A. ClusterProfiler was used to generate the functional interaction networks formed by upregulated genes related to the GO Biological Process terms. Dot size represents the number of genes in each GO term.

**Figure 5 jof-06-00311-f005:**
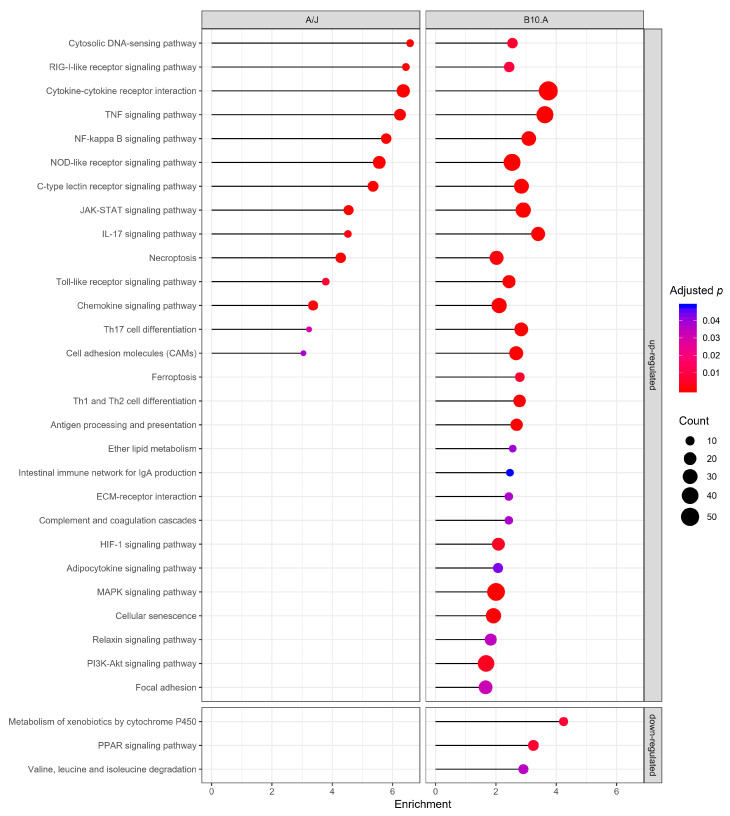
Kyoto Encyclopedia of Genes and Genomes (KEGG) pathway enrichment of differentially expressed genes in BMDCs derived from resistant (A/J) and susceptible (B10.A) mouse strains in response to *P. brasiliensis* infection. Enriched pathways (adjusted *p*-value < 0.05) associated with upregulated or downregulated DEG for the two murine strains. The *x*-axis represents the enrichment in each pathway. Dot size represents the number of genes in each pathway.

**Figure 6 jof-06-00311-f006:**
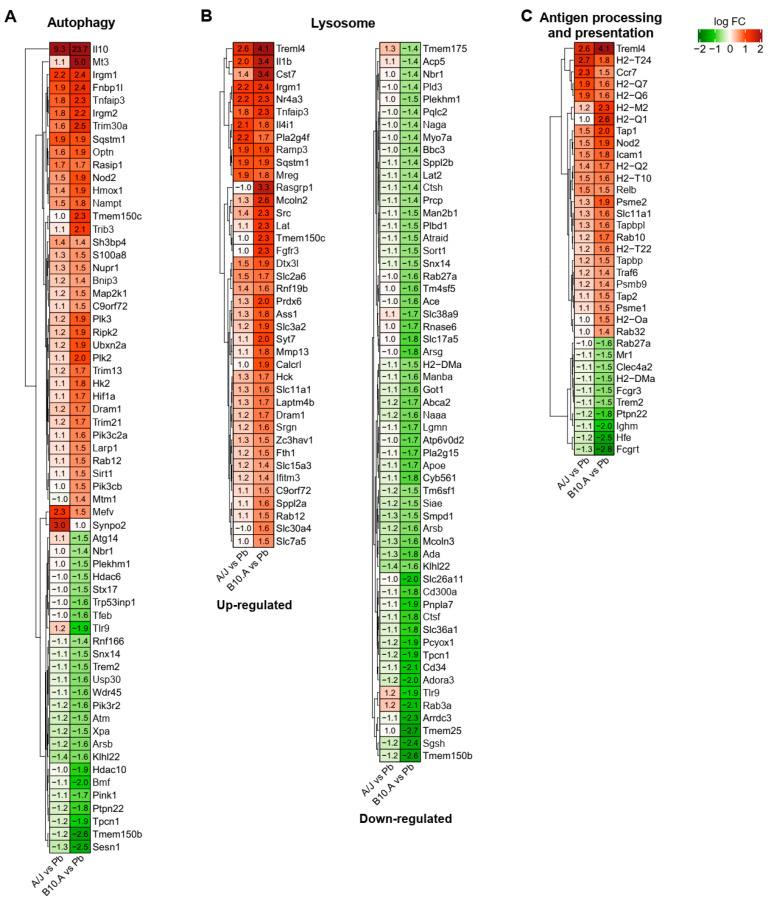
Heat map of differently expressed antigen presentation, autophagy and lysosome function genes (Gene Ontology terms) between BMDCs from resistant (A/J) and susceptible (B10.A) mouse strains upon infection with *P. brasiliensis*. (**A**–**C**) The values in heatmaps represent the fold change for each gene. DEGs: adjusted *p*-value < 0.05, and fold change ≥ ± 1.4.

**Figure 7 jof-06-00311-f007:**
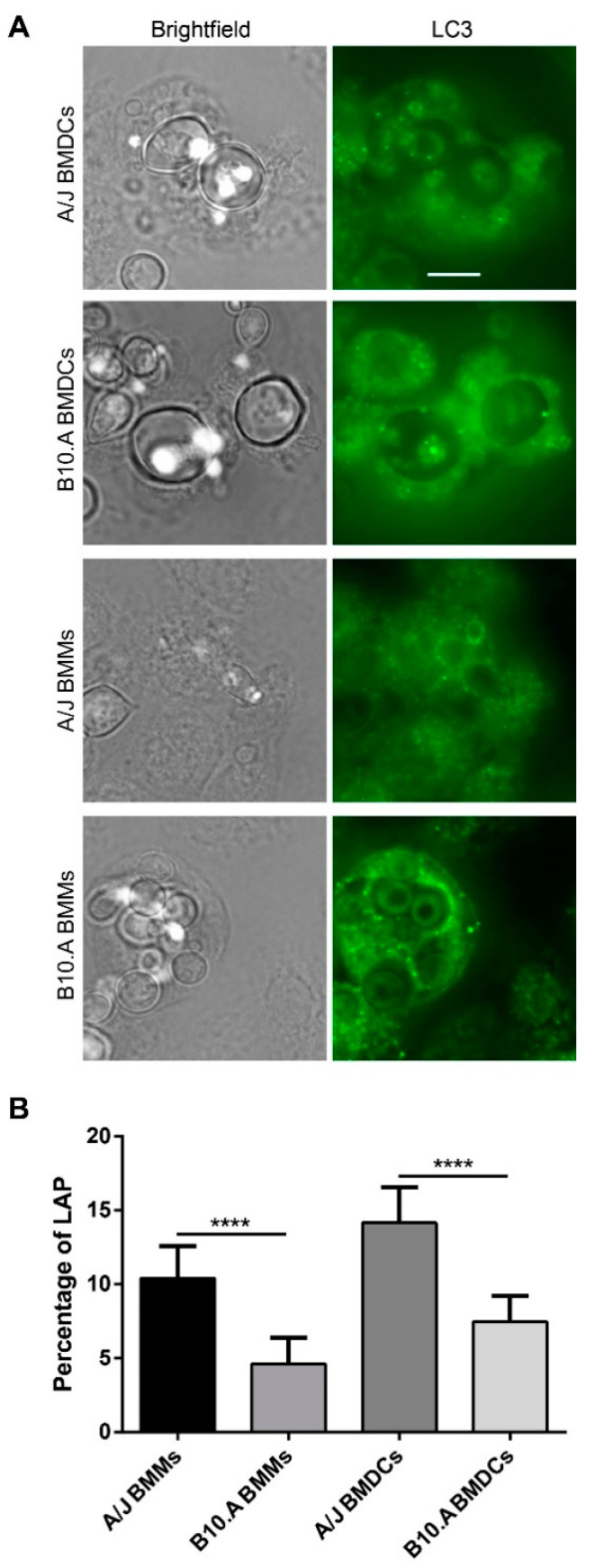
Light chain 3 (LC3)-associated phagocytosis of bone marrow-derived macrophages (BMMs) and BMDCs from A/J and B10.A mice upon *P*. *brasiliensis* infection. BMMs and BMDCs were cocultured with *P. brasiliensis* at a multiplicity of infection (MOI) 1:1 for 12 h and treated with anti-LC3 fluorescent antibody. (**A**) The localization of LC3-associated phagocytosis (LAP) was assessed by fluorescence microscopy. The fluorescence images were processed by deconvolution using a constrained iterative algorithm. The arrows point to cells that are surrounded by the autophagosome marker LC3. Scale bar: 10 µm. (**B**) The percentage of LC3-associated phagocytosis (LAP) was measured by the number of phagocytosed fungal cells positive for LC3 divided by the total number of phagocytosed fungal cells. Data are presented as mean ± 95% C.I. (*n* = 3 independent experiments, **** *p* < 0.0001 using Fisher’s exact test).

**Figure 8 jof-06-00311-f008:**
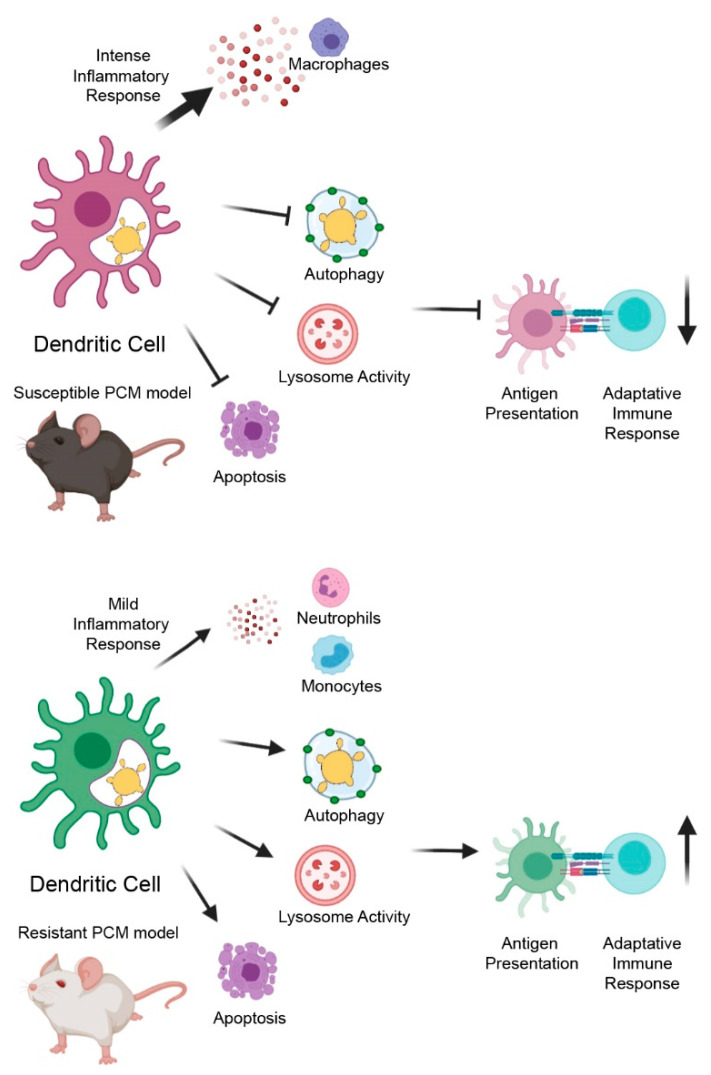
Schematic model of major transcriptional differences of BMDCs from resistant (A/J) and susceptible (B10.A) mouse strains upon infection with *P. brasiliensis*. BMDCs from the susceptible strain displayed a more intense activation of inflammatory response followed by the downregulation of autophagy, lysosome activity, and apoptosis, all processes involved in antigen processing and presentation and the activation of adaptive immune response. In contrast, BMDCs from the resistant mouse induce a mild inflammatory response with preserved functionality of autophagy, lysosome activation, and apoptosis, which might lead to more efficient antigen processing and presentation and proper activation of adaptative immune response.

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
