# Peer review of "Transcriptional Remodeling Patterns in Murine Dendritic Cells Infected with Paracoccidioides brasiliensis: More Is Not Necessarily Better"

_jof, 2020, doi:10.3390/jof6040311_

Round 1

Reviewer 1 Report

I would suggest adding the term “dendritic cells” to the article´s title

Authors should indicate the purity of DCs obtained from BM cultures (F480+ cells contamination)

Several question arise from results:

Which is the percentage of DCs engaged in yeast phagocytosis after ex-vivo infection at the MOI  used? Are resistant and susceptible DCs strains equivalent in this point? And their cell survival?

Before PCM infection, are DC´s transcriptomes from susceptible and resistant mice strains equivalent?  

Do DCs from resistant and susceptible mice strains reach a mature phenotype (costimulatory molecules , CCR7, MHC…) after ex-vivo PCM infection?  

Reviewer 2 Report

The manuscript describes PCM and its interaction with BMDC and transcriptional changes . The manuscript is well written. I have few comments as below

The introduction is well written but mortality level does not match (<5% vs <2% in abstract). I am not sure why authors has used macrophages in some experiments as entire manuscript is based on DCs . 

The authors did not show killing of PC in resistant vs susceptibility mice BMDCs

Fig 3 is missing in the manuscript

Fig 7 Can you please provide microscopy images with markers. What is the positive control

The transcripts may be put in the main manuscript. It would be great if the authors could confirm their RTPCR data with westerns and ELISA. 

The discussion is too long and need to more precise based on the results provided. 

A model of major transcriptional responses would be great as its difficult to read and understand the highlighted changes based on fig 4 and 5.

Author Response

The manuscript describes PCM and its interaction with BMDC and transcriptional changes. The manuscript is well written. I have few comments as below

1) The introduction is well written but mortality level does not match (<5% vs <2% in abstract). I am not sure why authors has used macrophages in some experiments as entire manuscript is based on DCs.

Answer: We agree with the reviewer and the discrepancy was corrected in text. The abstract was corrected to <5%. We only used macrophages in addition to dendritic cells in the experiments to validate the differences in modulation of autophagy which were suggested from the transcriptome analysis. We did so because macrophages are important effector cells in the immune response to P. brasiliensis.

2) The authors did not show killing of PC in resistant vs susceptibility mice BMDCs

Answer: We did not observe any difference in fungal viability at this time point. At 6 h of the ex vivo BMDC infection with P. brasiliensis, there was no significant reduction of fungal burden in either group.

3) Fig 3 is missing in the manuscript

Answer: We are sorry for this problem; to avoid this we are submitting the final version of the figures as separate files.

4) Fig 7 Can you please provide microscopy images with markers. What is the positive control?

Answer: We added representative pictures of LC3 recruitment in the different cells in figure 7 and we also added a positive control figure to the supplemental material based on our previously published work on macrophage LAP using other pathogenic fungi.

Nicola AM, Albuquerque P, Martinez LR, Dal-Rosso RA, Saylor C, De Jesus M, Nosanchuk JD, Casadevall A. Macrophage autophagy in immunity to Cryptococcus neoformans and Candida albicans. Infect Immun. 2012 Sep;80(9):3065-76. doi: 10.1128/IAI.00358-12. Epub 2012 Jun 18. PMID: 22710871; PMCID: PMC3418760.

5) The transcripts may be put in the main manuscript. It would be great if the authors could confirm their RTPCR data with westerns and ELISA.

Answer: We validated the modulation of selected transcripts using qPCR and of some cytokines by ELISA as presented in supplementary Figure S1 and S2, respectively. RT-qPCR is considered the golden standard for RNA-seq confirmation, and many genes modulated correspond to this model as described in the PCM resistance/susceptibility literature. Our focus was on the early transcriptome modulation and the qualitative differences between the two mouse strains. Unfortunately, due to the Covid-19 pandemic, the mouse models are not currently available for any further experiments.

6) The discussion is too long and need to more precise based on the results provided.

Answer: We agree with the reviewer and we reformulated the Discussion accordingly.

7) A model of major transcriptional responses would be great as its difficult to read and understand the highlighted changes based on fig 4 and 5.

Answer: We agree with the reviewer and we added a model which is presented as figure 8.

Reviewer 3 Report

Introduction

Please make sure that all the abbreviations are explained in the text.

An Abbreviation list is advisable, considering all the abbreviations used in the manuscript.

Materials and methods

For every commercial kit used please make sure that the name of the kit, manufacturer, city and country are mentioned.

2.12 Statistical analysis

Line 234: Tukey’s multiple comparison test, not “Turkey’s”

 Results

Figure 3 is missing.

Please move all the discussions from the Results section into Discussion section.

Discussion

Please list the strength and the limitations of the study.

Please indicate the clinical implications of the study.

Conclusions

Conclusions - please mention the clinical implications of the study.

Author Response

Reviewer 2

1) Introduction. Please make sure that all the abbreviations are explained in the text. An Abbreviation list is advisable, considering all the abbreviations used in the manuscript.

Answer: We have checked the text to make sure all the abbreviations are explained in their first appearance in the text.

2) Materials and methods. For every commercial kit used please make sure that the name of the kit, manufacturer, city and country are mentioned.

Answer: We agree with the reviewer and we have added the requested information.  

3) Statistical analysis. Line 234: Tukey’s multiple comparison test, not “Turkey’s”

Answer: We have corrected the typo.

4) Results.

4.1) Figure 3 is missing.

Answer: We are sorry for this problem; to avoid this we are submitting the final version of the figures as separate files.

4.2) Please move all the discussions from the Results section into Discussion section.

Answer: We agree with the reviewer and we reformulated the text from Results and from the Discussion accordingly.

  1. Discussion. Please list the strength and the limitations of the study. Please indicate the clinical implications of the study.
  2. Conclusions. Please mention the clinical implications of the study.

Answer: We believe that our work has to major strengths, to reveal some processes related to antigen presentation, such as autophagy, lysosome activity and apoptosis might be linked to poor activation of adaptive response to P. brasiliensis in the susceptible mouse model. In addition to that, we mentioned in several parts of our text how some of our results corroborate/confirm previously published results from other groups.

Regarding the limitations, the major one is that this is an in vitro study performed at only one time point of interaction. Hence, the whole immunological context (other cells, signals, and interactions) is not considered. Further in-depth exploration and assessment of these processes in vitro and in vivo should help deepen the comprehension of the molecular mechanisms behind susceptibility/resistance to this neglected fungal infection.

Thus, we reformulated the conclusion paragraph to address these aspects and possible clinical implications of our results as presented below:

“In conclusion, our results corroborate the previously proposed intense activation of the inflammatory response in the susceptible PCM mouse model after infection with P. brasiliensis. In addition to that, we propose that suppression of highly interconnected processes, such as repression of lysosomal acidification, catalytic activity, and autophagy function, might negatively impact antigen processing and presentation by BMDCs from the susceptible mice leading to ineffective activation of the adaptive immune response and susceptibility to this fungal infection (Figure 8). What makes a host susceptible or resistant to infection is a crucial question for most infectious diseases, and many factors have been implicated in disease development, such as sex, nutritional status, smoking habits, pollution, and genetics [89]. Our work reinforces the significance of host genetic background in susceptibility to fungal infections. These findings might help develop strategies to prevent or treat PCM, such as the use of DC biomarker detection to predict people with higher risks of developing the disease, information with significant prognostic impact. Therefore, further in-depth exploration and assessment of these processes, for example, in vivo, with other cells or different time points of interaction, should help deepen the comprehension of the molecular mechanisms behind susceptibility/resistance not only for this neglected fungal infection but also for other infectious diseases.”

Round 2

Reviewer 2 Report

No comments